

# Expression and characterization of thermotolerant lipase with broad pH profiles isolated from an Antarctic *Pseudomonas* sp strain AMS3

Wahhida Latip[1,*], Raja Noor Zaliha Raja Abd Rahman[1,2], Adam Thean Chor Leow[1,3,*], Fairolniza Mohd Shariff[1,2,*] and Mohd Shukuri Mohamad Ali[1,4,*]

[1] Enzyme and Microbial Technology Research Center, Universiti Putra Malaysia, Serdang, Selangor, Malaysia

[2] Department of Microbiology, Faculty of Biotechnology and Biomolecular Sciences, Universiti Putra Malaysia, Serdang, Selangor, Malaysia

[3] Department of Cell and Molecular Biology, Faculty of Biotechnology and Biomolecular Sciences, Universiti Putra Malaysia, UPM Serdang, Selangor, Malaysia

[4] Department of Biochemistry, Faculty of Biotechnology and Biomolecular Sciences, Universiti Putra Malaysia, Serdang, Selangor, Malaysia

[*] These authors contributed equally to this work.

## ABSTRACT

A gene encoding a thermotolerant lipase with broad pH was isolated from an Antarctic *Pseudomonas* strain AMS3. The recombinant lipase AMS3 was purified by single-step purification using affinity chromatography, yielding a purification fold of approximately 1.52 and a recovery of 50%. The molecular weight was approximately ∼60 kDa including the strep and affinity tags. Interestingly, the purified Antarctic AMS3 lipase exhibited broad temperature profile from 10–70 °C and stable over a broad pH range from 5.0 to pH 10.0. Various mono and divalent metal ions increased the activity of the AMS3 lipase, but $Ni^{2+}$ decreased its activity. The purified lipase exhibited the highest activity in the presence of sunflower oil. In addition, the enzyme activity in 25% v/v solvents at 50 °C particularly to n-hexane, DMSO and methanol could be useful for catalysis reaction in organic solvent and at broad temperature.

Corresponding author
Raja Noor Zaliha Raja Abd Rahman, rnzaliha@upm.edu.my

## INTRODUCTION

Psychrophiles produce cold active enzymes, which are more active at low and moderate temperatures compared to homologous mesophilic enzymes. These enzymes are capable of maintaining the flexibility and dynamics of their active site at low temperatures. The enzymes are highly useful in biotechnological application as they save energy costs, prevent undesired chemical transformation, prevent the loss of volatile compounds and do not require expensive heating or cooling systems (*Margesin et al., 2003*).

Lipases or triacylglycerol acylhydrolases (EC 3.1.1.3) are long chain fatty acid ester hydrolases and are currently attracting enormous attention because of their biotechnological potential. Lipases occur widely in nature and are largely produced by animals, plants and

microbes; microbial lipases are the most studied and commercialized. Microbial lipases have gained huge attention from industry because of their flexibility in temperature, pH and organic solvent (*Verma, Thakur & Bhatt, 2012*). Lipases have been widely used for biotechnological and industrial applications such as in the food and leather industries, oil processing, environmental management and the production of surfactants, detergents, and pesticides. The temperature stability of lipases has been regarded as the most important characteristic for use in industry. Cold-adapted lipases are largely used in the detergent industry, where cold washing reduces both energy consumption and the wear and tear of textile fibers. These lipases are also preferred in the food industry, as these enzymes can be inactivated at reasonably low temperatures, thereby conserving the nutritional quality of the food. Lipase specificity to catalyze triglycerides in particular has attracted application in nutraceutical industry. Specific structured triacylglycerols (SSTs) (*Xuebing, 2000*) is a product mainly produced via lipase reactions in which the chemical catalysts are unable to do. They are also used in environmental applications such as wastewater treatment and the bioremediation of fat-contaminated cold environments (*Lu et al., 2010*).

Chemical processes, including biomass conversion, bioremediation and fermentation, are highly desired to be completed at lower temperatures because this approach reduces the risk of contamination by mesophiles and saves on energy consumption (*Cotarlet & Bahrim, 2011*). In conjunction with these needs, studies focused on enzymes with unique properties (i.e., broad temperature and pH profiles) have gained an increasing interest. To the best of our knowledge, there is no report pertaining to Antarctic lipases exhibiting activities at broad temperature and pH ranges. This current work describes the gene isolation, expression, recombinant lipase purification and characterization of a lipase from Antarctic *Pseudomonas* with broad temperature and pH profiles.

## MATERIALS AND METHODS

### Sources of bacteria

The bacterium was isolated from Antarctic soil. The isolated denoted as AMS3. Identification was performed using 16SrDNA and homology to *Pseudomonas* spp. The 16SrDNA sequence was sent to NCBI under accession no KR821141. The isolate was screened to produce lipase via qualitative approach using selective media tributyrin, Rhodamine B and Victoria Blue agar plates containing tributyrin, triolen and olive oil respectively as substrate (*Samad et al., 1989*). Lipase producer will hydrolyze the lipid to free fatty acid. Lipolysis is observed directly by changes in the appearance of the substrate such as forming a clearing zone and change in colour of indicator dye used (*Scholze et al., 1999*).

### Quantitative assay for lipase activity

The lipase assay was performed with a colorimetric method using olive oil as the substrate as previously described by *Kwon & Rhee (1986)*. A reaction mixture of 1.0 ml of enzyme, 2.5 ml olive oil emulsion (50% olive oil +50% phosphate buffer, the emulsion was mix using homogenizer at 2,500 rpm) and 0.02 ml $CaCl_2.2H_2O$ was used. The reaction mixture was incubated for 30 min with shaking (250 rpm) at 37 °C. The reaction was stopped by the addition of 5.0 ml isooctane. The upper layer (4.0 ml) was moved to a test tube, and

1.0 ml of cupric acetate pyridine pH 6.1 was added. The concentration of free fatty acid dissolved in isooctane was determined by measuring the absorbence at 715 nm. One unit of lipase activity is defined as the rate of release one micromole of free fatty acid in one min.

## Cloning of the lipase gene
### Genomic library construction
*Pseudomonas* sp strain AMS3 genomic DNA was extracted using the Qiagen DNA Extraction kit (Qiagen, Hilden, Germany). Partial digestion of *Pseudomonas* AMS3 genomic DNA was performed using *Sau*3A1 (Fermentas, Germany) to prepare a clonable size of genomic DNA. The plasmid was digested with *Bam*H1 to generate compatible ends for ligation with *Sau*3A1 partially digested genomic DNA. The plasmid DNA (50 µl) was digested with *Bam*H1 for 1 h in a 100 µl reaction mixture that included the following components: (1) distilled water (37 µl), (2) 10X buffer E (10 mM Tris–HCl, pH 7.4, 300 mM KCl, and 5 mM MgCl$_2$,) (3) 0.1 mM Dithiothreitol (DTT) (10 µl), (40 BSA (0.1 µg;1 µl), (5) DNA (50 µg), and (6) enzyme (10U; 2 µl). The mixture was incubated at 37 °C for 1 h. The cleaved products were dephosphorylated by the direct addition of calf intestinal alkaline phosphatase (1 U, 1 µl) for 1 h at 37 °C. The enzyme was deactivated by heating at 65 °C for 20 min in the presence of EDTA (5 mM). The *Sau*3A1 partially digested genomic DNAs with maximum sizes in the range of 2–9 kb were ligated with the pET 51b vector (*Bam*H1 digested and dephosphorylated).

### Screening for positive recombinant clones
Putative lipase activity was indicated by the formation of halos around the colonies on tributyrin-ampicillin agar. Cold active lipase activity was confirmed by incubating the plates at 37 °C for 16 h followed by another 72 h of incubation at 4 °C. Colonies that produced clearing zones on tributyrin-ampicillin agar were isolated and restreaked on Victoria Blue agar and Rhodamine B agar plates to confirm lipase activity.

### Gene sequencing
Plasmid from positive transformant was extracted and digested with the BamH1. The insert was predicted around 9 kb. The T7 promoter and terminator was use to sequence the gene via primer walking. Result from the starting primer was sent to 1st BASE Laboratories Sdn Bhd (Shah Alam, Selangor, Malaysia) for primer walking. The primer was designed base on each sequencing result.

### Analysis of the lipase gene
Lipase sequence analysis was performed using the National Centre of Biotechnology (NCBI) database. Analysis of the lipase gene was performed using Biology Workbench (http://workbench.sdsc.edu) and the Expasy Molecular Biology Server (http://www.expasy.com/tools). The nucleotide sequence of the lipase gene has been submitted to the Genbank database with the accession number KP744536 and KT215165.

### Expression of recombinant lipase in E. coli
A cultured derived from a colony harboring the putative lipase gene (*E. coli BL21*/pETlipAMS3) was cultured in LB media containing 50 mg/L ampicillin. A total of

200 ml of LB media was inoculated with transformed cells or empty vector and incubated at 37 °C with shaking at 150 rpm. A total of 0.5 mM of IPTG was added to cultures with an $A_{600}$ of 0.5–0.7 and the cells were grown at 20 °C for 8 h. Cell growth was measured spectrophotometrically at $A_{600}$.

## Recombinant AMS3 lipase purification

The His-tagged recombinant AMS3 lipase was purified using Nickel Sepharose affinity chromatography. The cell pellet was resuspended using 100 mM Tris–HCl (pH 7.4) binding buffer containing 20 mM imidazole and 0.5 M NaCl. The sonicated cells were centrifuged at $12,000 \times g$ for 30 min to separate the crude extract. The crude enzyme was filtered and subjected to affinity columns that were pre-equilibrated with binding buffer. The recombinant enzyme was eluted using elution buffer containing 100 mM Tris–HCl with 0.5 M imidazole and 0.5 M NaCl (pH 7.4). The active fractions were pooled together and the protein homogeneity was determined using SDS-PAGE.

## Protein determination

The protein content was measured by the Bradford method (*Bradford, 1976*); BSA was used as a standard. During chromatography, the protein concentrations in the fractions were monitored by measuring the absorbance at 280 nm.

## SDS-PAGE analysis

The purified fractions that showed lipase activity were pooled and subjected to SDS-PAGE according to the method described by *Laemmli (1970)*. The sample was electrophoresed on a 12% resolving gel (2.5 ml 1.5 M Tris–HCl (pH 8.8), 100 µl 10% SDS, 4.4 ml distilled water, 3 ml Bis-acrylamide (12%), 50 µl APS and 10 µl TEMED) and a 6% stacking gel (2.5 ml 0.5 M Tris–HCl (pH 6.8), 100 µl 10% SDS, 5.9 ml distilled water, 1.5 ml 12% Bis-acrylamide and 50 µl APS). Electrophoresis was performed at a constant voltage (210 V) at room temperature. Coomassie Brilliant Blue R 250 was used to stain the gel for 10 min, and then the gel was destained with destaining solution. The molecular mass of the protein was estimated using a protein standard marker (Unstained protein marker from Thermo scientific USA).

## Characterization of AMS3 lipase
### Effect of temperature

The effect of temperature on the recombinant lipase was studied by measuring the enzyme activity at varying temperatures (5 °C, 10 °C, 20 °C, 30 °C, 40 °C, 50 °C, 60 °C and 70 °C) for 30 min at pH 7. The activity was assayed following the method of *Kwon & Rhee (1986)* and the assay was done using olive oil as substrate. To assess the effect of temperature on the thermostability of the lipase, the enzyme solution was pre-incubated in 50 mM Phosphate buffer (pH 7) at different temperatures (5 °C, 10 °C, 20 °C, 30 °C, 40 °C, 50 °C, 60 °C and 70 °C) for 30 min and assayed at the optimum temperature.

## Effect of pH

The optimum pH of the recombinant lipase was studied by emulsifying the substrate (olive oil) in buffers of different pH values (4.0–12.0). The following buffers (50 mM) were used:

(1) sodium acetate (pH 4.0–6.0), (2) potassium phosphate (pH 6.0–8.0), (3) Tris–HCl (pH 8.0–9.0) and (4) glycine-NaOH (pH 9.0–12.0). To assess the pH stability of the lipase, the enzyme solution was pre-incubated in different pH buffers (pH 4.0–12.0) at 50 °C for 30 min, followed by the lipase assay.

## Effect of metal ions

The effect of metal ions on lipase activity was studied using the method described by *Kwon & Rhee (1986)* after the enzyme was pre-incubated at 50 °C for 30 min in 50 mM Phosphate buffer (pH 8.0) and either 1 mM or 5 mM of various metal ions in different reaction vessels ($Li^+$, $Rb^+$, $Na^+$, $Mg^{2+}$, $Ca^{2+}$, $Fe^{2+}$, $Mn^{2+}$, $K^+$, $Zn^{2+}$, $Ni^{2+}$ or $Co^{2+}$).

### Natural oil specificity

Natural oils (olive oil, corn oil, sun flower oil, canola oil, sesame oil, coconut oil and palm oil) were tested for AMS3 lipase activity. The oils were assayed at 50 °C for 30 min colorimetrically.

### Effect of organic solvents

The effects of organic solvents on lipase activity were determined by measuring the residual activity after pre-incubation of the enzyme with different solvents at a concentration of 25% (v/v) in 50 mM Phosphate buffer (pH 8.0) at 50 °C for 30 min. The residual activity was measured following the method of *Kwon & Rhee (1986)* and using olive oil as a substrate. The solvents used had varying *log P* values as shown in parenthesis. The following solvents were used: (1) DMSO (−1.45), (2) methanol (−0.76), (3) acetonitrile (−0.33), (4) ethanol (−0.24), (5) acetone (−0.24), propanol (0.28), (6) chloroform (2.0), (7) benzene (2.0), (8) toluene (2.5), (9) xylene (3.1) and (10) n-hexane (3.5).

### Statistical analysis

The standard deviations of the triplicate data were performed using deviation (SD) in Microsoft office excel 2010 (Microsoft Corporation USA). The data are mean ±standard deviation of three determinations and indicated as error bars. When the error bar cannot see seen, they are less than the size of symbol.

## Secondary structure and thermal denaturation measurement of AMS3 lipase using Circular dichroism (CD) spectropolarimeter

Circular dichroism (CD) spectra were recorded using JASCO J-810 spectropolarimeter at 25 °C. The purified AMS3 lipase was dialysed over night with 10 mM phosphate buffer pH 7 prior to CD spectral analysis. The secondary content measurement was conducted from wavelength of 190 to 260 nm on a 1 mm path length. A few temperatures have been set to measure the changes of secondary structure from 10 °C to 90 °C. The protein concentration was 0.1 mg/ml and the cell pathlength 0.1 cm. The data been collected every 1 nm (band with) and the data pitch every 0.5 nm. Protein secondary structures content were estimated from the far-UV CD spectra based on the following link: http://perry.freeshell.org/raussens.html (*Raussens, Ruysschaert & Goormaghtigh, 2003*).

The thermal denaturation of AMS3 lipase was measured at 222 nm from 10 °C to 90 °C at a 1 °C/min heating rate. Wavelengths 222 nm measures $\alpha$-helical content of the protein.

Data collected from higher wavelengths usually have lower absorption. $T_m$ is defined as a midpoint of sigmoidal melting curves using 0.5 mg/ml protein. The data was collected every 1 degree per min. Data pitch, bandwidth, response, scanning speed, and accumulation were set to be 0.1 degree, 1 nm, 1 s, 1 degree per min and 3 times, respectively.

## RESULTS AND DISCUSSION

### Lipase gene isolation and expression in *E. coli*

To isolate the lipase gene, a genomic library approach was completed. To achieve this, a library was constructed via fragmentation or partial digestion of genomic DNA using *Sau*3A1. The plasmid (pET51b) was linearized using *Bam*HI. Among thousands of transformants from the shotgun cloning, three of them produced a clearing zone. The positive recombinants harboring the putative lipase gene were assumed to have cold active lipolytic activity when the recombinant clone was initially grown at 37 °C for 24 h and transferred at 4 °C. The clearing zone only appeared after 3 days of incubation at 4 °C. The colonies were transferred to two selective media plates, triolein and Rhodamine B plates, to confirm lipase gene expression. Colonies 1 and 2 gave a positive result by the intense blue and orange coloration around the colonies. Colony 3, however, did not exhibit such coloration. These results show that these two colonies produced a putative true lipase compared to colony 3 that was a putative esterase producer (data shown in Data S1). To identify a true lipase producer the transformant must able to hydrolyse triglyceride that contain long chain fatty acid (*Scholze et al., 1999*). Colony 1 and 2 were sent for sequencing and the partial sequence from both colonies shown to contain putative gene from $\alpha/\beta$ hydrolase family. Colony 1 (denoted as AMS3 lipase) was selected due to the high catalytic activity at 20 °C and was used for the remainder of the study.

Generally, protein functional and structural studies often require a large amount of pure, correctly folded protein, which is commonly produced in *Escherichia coli* expression systems. However, these overexpressed proteins are sometimes not efficiently processed by the *E. coli* post-translational machinery, resulting in protein misfolding. A dense insoluble aggregate of misfolded proteins is generally known as inclusion bodies (IBs). Most of the lipases from *Pseudomonas* are expressed in the form of IBs. Even under the control of pET-25b(+), a vector containing a signal sequence responsible for the translocation of expressed protein to the periplasm, the lipases from *Pseudomonas* sp. MIS38 and *Pseudomonas* sp. strain KB700A were still overexpressed in non-active, insoluble forms (*Claudia et al., 2009*; *Rashid, Shimada & Ezaki, 2001*).

### Lipase sequence analysis

Putative plasmid contained recombinant lipase AMS3 was sent for sequencing. A few sets of primer were used to complete the sequence. The presence for open reading frame (ORFs) was analyzed using the ORF finder software from the NCBI. One open reading frame (ORF) was found. With a predicted size of 1353 bp encoding a 450 amino acid was determined. The protein sequence is made up by two different domains namely the GST C (KT215165) domain and lipase (KP744536) domain at the N and C terminal sections respectively. The lipase (AMS3 lipase) molecular mass without the GST C sequence was

predicted to be 43.6 kDa with pI value of 6.3. The BLASTp result revealed that AMS3 lipase showed homology to $\alpha/\beta$ hydrolase family. Multiple sequence alignment was done using Biology Workbench software to predict the conserve region of the lipase. Aligment between known lipase from various microorganisms exhibited the conserved region of the lipase (Fig. 1). A typical lipase conserved region contains the pentapeptide (GXSXG) sequence. The lipase catalytic triad which made up by Ser, Asp and His were also conserved. No signal peptide sequence was observed as predicted using SignalP software. Similarly, Group III lipases including lipases from *Pseudomonas fluorescens* and *Serratia marcescens* has no signal peptide. The protein origin can be reveal from G + C content. The G + C content was calculated using http://watson.nih.go.jp (FramePlot 2.3.2). AMS3 lipase content 55% of G + C almost similar to lipase from *Pseudomonas fragi* (59.3%) (*Claudia et al., 2009*). The N terminal section of AMS3 lipase contains a GST C (glutathione-S-transferase C family) similar to *Pseudomonas* Ag1. Glutathione have been reported as essential factor for chaperon to activate the lipase from *Pseudomonas* (*Rosenau & Jaeger, 2000*; *Tanaka, Nihira & Yamada, 2000*). According to Tanaka and co-workers, glutathione was found to be important for 'lipase activator factor' or *Lif* protein from *Pseudomonas* sp strain 109 to fold an active form (*Tanaka, Nihira & Yamada, 2000*).

## Purification of AMS3 lipase

*E. coli* BL21 cells (DE3) harboring the pET51b/lipAMS3 plasmid was used to overexpress the recombinant AMS3 lipase under the control of a T7 promoter. The crude AMS3 lipase containing 10x histidine and Strep tags was produced in a soluble form allowing for rapid purification of the recombinant protein. The addition of fusion affinity tags and GST C *Pseudomonas* A3 increased the size of AMS3 lipase to ∼60 kDa. Purification was accomplished using the XK16 affinity chromatography column (GE Healthcare, USA) packed with Nickel Sepharose resin. After the column was equilibrated with binding buffer, the crude sample was loaded onto the column. Elution was accomplished using increasing concentrations of imidazole. Pooled fractions resulted in a 50% yield. Some protein was retained in the low-yield fractions and not included in the pooled fraction, resulting in some loss. Pooled fractions had an overall purification factor of approximately 1.52 fold. In 2007, *Yan et al. (2007)* reported that their organic solvent-stable and thermostable lipase from *Galactomyces geotrichum* Y05 had up to a 32% recovery and a 3.2-fold purification factor using ammonium sulfate, an ion exchange column and gel filtration purification. SDS-PAGE analysis indicated that the molecular weight of the AMS3 lipase was approximately ∼60 kDa (Fig. 2). Generally, lipases have molecular weights in the range of 16–69 kDa. According to *Schmidt-Dannert (1999)*, the *Staphylococcal* lipase family, which includes lipase BTL-2 from *B. thermocatenulatus*, possesses molecular weights of 40–45 kDa (*Schmidt-Dannert, 1999*).

## Biochemical characteristics of AMS3 lipase
### Effect of temperature

Lipase activity was determined from 5 to 70 °C. The optimum temperature for the lipase was activity was at 50 °C, and the activity dropped above 60 °C (Fig. 3A). Interestingly,

```
AMS3_lipase             ------------------------------------------------ASLRA
chain_A_t1              -------------------------------------------------SLRA
pseudomonas_fragi       ------------------------------------------------MDDSVN
pseudomonas_arg_Q9L6C7  ------------------MKKKSLLPLGLAIGLASLAASPLIQASTYTQ
proteus_sp_k107         ---------------------------------------------------MS
pseudomonas_flo         METGIFDYKNLGTEGSKTLFADAMETAITLYSYHNLDNGFAVGYQHNGLG

AMS3_lipase             NDAPIVLLHGFTGWGREE--MFGFKYWGGVRGDIEQWLNDNGYRTYTLAV
chain_A_t1              NDAPIVLLHGFTGWGREE--MFGFKYWGGVRGDIEQWLNDNGYRTYTLAV
pseudomonas_fragi       TRYPILLVHGLFGFD--R--IGSHHYFHGIK----QALNECGASVFVPII
pseudomonas_arg_Q9L6C7  TKYPIVLAHGMLGFD--N--ILGVDYWFGIP----SALRRDGAQVYVTEV
proteus_sp_k107         TKYPIVLVHGLAGFN--E--IVGFPYFYGIA----DALRQDGHQVFTASL
pseudomonas_flo         LGLPATLVGALLGSTDSQGVIPGIPWNPDSEKAALEAVQKAGWTPISASA
                         *    *   .: *      .  : . :  .       . :.  *

AMS3_lipase             GPLSSNWDRACEAYAQLVGGTVDYGAAHAAKHGHARFGRTYPGLLPELKR
chain_A_t1              GPLSSNWDRACEAYAQLVGGTVDYGAAHAAKHGHARFGRTYPGLLPELKR
pseudomonas_fragi       SAANDNEARGDQLLKQIHN------------------------LRRQVG
pseudomonas_arg_Q9L6C7  SQLDTSEVRGEQLLQQVEE------------------------IVALSG
proteus_sp_k107         SAFNSNEVRGKQLWQFVQT------------------------LLQETQ
pseudomonas_flo         LGYAGKVDARGTFFGEKAGYTTAQVEVLGKYDDAGKLLEIGIG--FRGTS
                                   .

AMS3_lipase             GGRIHIIAHSQGGQTARMLVSLLENGSQEEREYAKAHNVSLSPLFEGGHH
chain_A_t1              GGRIHIIAHSQGGQTARMLVSLLENGSQEEREYAKAHNVSLSPLFEGGHH
pseudomonas_fragi       AQRVNLIGHSQGALTARYVAAIAP-------------------------E
pseudomonas_arg_Q9L6C7  QPKVNLIGHSHGGPTIRYVAAVRP-------------------------D
proteus_sp_k107         AKKVNFIGHSQGPLACRYVAANYP-------------------------D
pseudomonas_flo         GPRETLISDSIGDLISDLLAALGPKDYAKNYAGEAFGGLLKNVADYAGAH
                         :   :*..* *         :.:                           .

AMS3_lipase             FVLSVTTIATPHDGT--------------------TLVNMVDFTDRFFD
chain_A_t1              FVLSVTTIATPHDGT-------------------TLVNMVDFTDRFFD
pseudomonas_fragi       LIASVTSVSGPNHGS--------------------ELADRLRLAFVPGR
pseudomonas_arg_Q9L6C7  LIASATSVGAPHKGS--------------------DTADFLR-QIPPGS
proteus_sp_k107         SVASVTSINGVNHGS--------------------EIADLYRRIMRKDS
pseudomonas_flo         GLTGKDVVVSGHSLGGLAVNSMETADLSNYKWAGFYKDANYVAYASPTQS
                         : .   :   :                                   .:

AMS3_lipase             LQKAVLEAAAVASNVPYTSQVYDFKLDQWGLRRQPGE-SFDHYFERLKRS
chain_A_t1              LQKAVLEAAAVASNVPYTSQVYDFKLDQWGLRRQPGE-SFDHYFERLKRS
pseudomonas_fragi       LGETVAAALTTSFSAFLSALSGHPRLPQNALNALNAL-TTDGVAAFNRQY
pseudomonas_arg_Q9L6C7  AGEAILSGLVNSLGALISFLSSGSTGTQNSLGSLESL-NSEGAARFNAKY
proteus_sp_k107         IPEYIVEKVLNAFGTIISTFSGHRGDPQDAIAALESL-TTEQVTEFNNKY
pseudomonas_flo         AGDKVLNIGYENDPVFRALDGSSFNLSSLGVHDKPHESTTDNIVSFNDHY
                         . :        . :          . .:       . :        :

AMS3_lipase             PVWTSTDTARYDLSVSGAEKLNQWVQASPNTYYLSFSTERTYRGALTGNH
chain_A_t1              PVWTSTDTARYDLSVSGAEKLNQWVQASPNTYYLSFSTERTYRGALTGNH
pseudomonas_fragi       PQGLPDRWG-----GMGPAQVN-------AVHYYSWS------GIIKGSR
pseudomonas_arg_Q9L6C7  PQGIPTSAC-----GEGAYKVN-------GVSYYSWS----------GS-
proteus_sp_k107         PQALPKTPG-----GEGDEIVN-------GVHYYCFG------SYIQGLI
pseudomonas_flo         ASTLWNVLP------FSIVNLPTWVSHLPTAYGDGMETTRILESGFYDQM
                         .           .       :          .           .
```

**Figure 1  Alignment of the AMS3 lipase sequence with *Pseudomonas* lipases of known three-dimensional structure.** The AMS3 was aligned with *P. aeruginosa* lipase (Q9L6C7), *P. fragi* lipase (AY787823), *G. bacillus* T1 (AY166603) and *Proteus* sp. K107. −, Serine Pentapeptide; @, Aspartic acid; and ,ˆ Amino acid involve in calcium binding. (*) blue: Conserved residue; (:) green: conservation of strong groups; (.) blue: conservation of weak groups.

```
                                                                        @
AMS3_lipase          YPELGMNAFSAVVCAPFLGSYRNPTLGIDDRWLENDGIVNTVSMNGPKRG
chain_A_t1           YPELGMNAFSAVVCAPFLGSYRNPTLGIDDRWLENDGIVNTVSMNGPKRG
pseudomonas_fragi    LAESLN------LLDPLHNALRVFDSFFTRETRENDGMVGRFSSH-----
pseudomonas_arg_Q9L6C7 -SPLTN------FLDPSDAFLGASSLTFKNGT-ANDGLVGTCSSH-----
proteus_sp_k107      AGEKGN------LLDPTHAAMRVLNTFFTEKQ--NDGLVGRSSMR-----
pseudomonas_flo      ETTRDSTVIVANLSDPARANTWVQDLNRNAEPHKGNTFIIGSDGNDLIQG
                               .    *                     .: ::    . .
                                                  ^
AMS3_lipase          SSDRIVPYDGTLKKGVWNDMGTYNVDHLEIIG----VDPNPSFDIRAFYL
chain_A_t1           SSDRIVPYDGTLKKGVWNDMGTYNVDHLEIIG----VDPNPSFDIRAFYL
pseudomonas_fragi    -----------LGQVIRSDYPLDHLDTINHMA----RGSRRRINPVELYI
pseudomonas_arg_Q9L6C7 -----------LGMVIRDNYRMNHLDQVNQVFG---LTSLFETSPVSVYR
proteus_sp_k107      -----------LGKLIKDDYAQDHIDMVNQVAG---LVG-YNEDIVAIYT
pseudomonas_flo      GNGADFIEGGKGNDTIRDNSGHNTFLFSGHFGNDRVIGYQPTDKLVFKDV
                               : .:      .      .           .
AMS3_lipase          RLAEQLASLRP---------------------------------
chain_A_t1           RLAEQLASLQP---------------------------------
pseudomonas_fragi    EHAKRLKEAGL---------------------------------
pseudomonas_arg_Q9L6C7 QHANRLKNASL---------------------------------
proteus_sp_k107      QHAKYLASKQL---------------------------------
pseudomonas_flo      QGSTDLRDHAKVVGADTVLTFGADSVTLVGVGHGGLWTEGVVIG
                     . :   *  .
```

**Figure 1 (…continued)**

the purified lipase exhibited a broad temperature profile, whereby the lipase was still active at temperatures ranging from 10–60 °C. This finding is unlike the typical characteristics of cold active lipase. Most cold active lipases have been found to be active at temperature ranges from 15–35 °C (*Gerday et al., 2000*). However, few have reported the broad temperature observation from psychrophilic/psychrotolerant bacteria. A *Pseudomonas* lipase from Greenland exhibited an almost similar activity at wide temperatures (*Schmidt, Larsen & Stougaard, 2010*). In contrast, AMS3 lipase exhibited higher optimum activity compared to other reported lipases isolated from psychrophilic bacteria. r-*LipA* from *Sorangium cellulosum* (*Cheng et al., 2011*), *lipA1* from *Psychrobacter* sp. 7195 (*Zhang et al., 2009*), lipase from *Pseudomonas fragi* strain IFO 3458 (PFL) (*Alquati et al., 2002*) and *lipX* from *Psychrobacter* sp. C18 (Alquati et al. 2011) were all reported to show maximum activity at 30 °C. AMS3 lipase was mostly stable at 30 °C and below (Fig. 3B). The organic solvent-stable and thermostable lipase from *Galactomyces geotrichum* Y05 was stable below 60 °C at a pH of 8.0 for at least 12 h; however, above 60 °C, rapid inactivation occurred with only 36% residual activity. These characteristics differ from those of the AMS3 lipase and may be due to the effects of the nature of the enzyme itself, which originated from Antarctica.

### Effect of pH

The broad temperature range of AMS3 lipase showed activity in the pH range from 5.0 to 10.0; the optimum activity was seen at pH 8.0 (Fig. 4A). Lipase activity was reduced above a pH of 8.0, with a relative activity of 34% at pH 9.0. Most of the studied *Pseudomonas* lipases have been known to be active around pH 7 to 8, stable at pH ranging from 6 to 9. To date, only a few lipases reported an optimum activity at pH 9 to 10. The cold-adapted lipase from *Psychrobacter* sp. 7195 (*Zhang et al., 2007*) and a mesophilic lipase from *Pseudomonas*

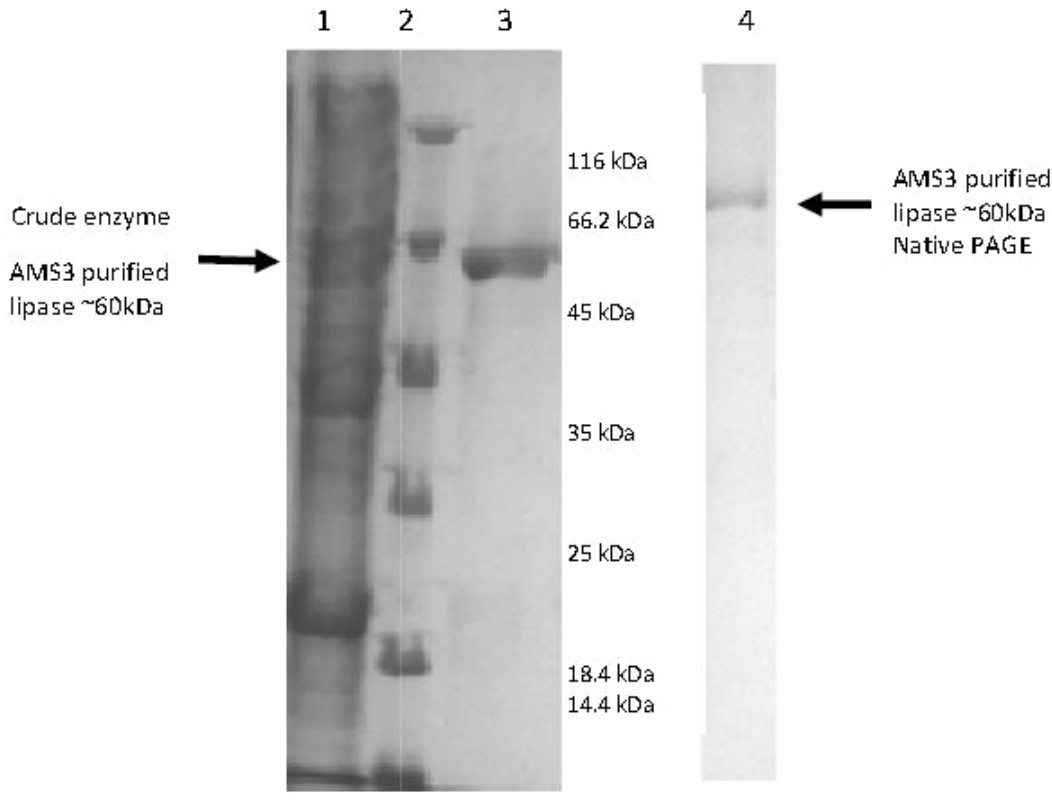

**Figure 2** **Polyacrylamide gel electrophoresis (PAGE) profile of recombinant AMS3 lipase.** Lane; Lane 1: Crude lipase; Lane 2: Protein molecular weight marker(Fermentas, Germany); Lane 3: Purified AMS3 lipase; Homogeneity of AMS3 lipase was determined via native PAGE (Lane 4). The mobility of the purified lipase was estimated around ~60 kDa (indicated by arrow). The protein loaded was approximately around 13 µg.

*fluorescens* JCM5963 (*Zhang et al., 2009*) reported an optimum activity at pH 9, "whereas" lipases from *Acinetobacter* sp. XMZ-26 (cold-adapted) (*Zheng et al., 2011a*; *Zheng et al., 2011b*) and *Pseudomonas fluorescens* AK102 (*Kojima, Kobayashi & Shimizu, 2003*) reported an optimum activity at pH 10. The AMS3 lipase was stable from pH 5.0 to 10.0 after incubation for 30 min at 50 °C (Fig. 4B). Extracellular lipase from *Thermomyces lanuginosus* was also reported to be stable at in broad range alkaline pH from pH 8–12 (*Zheng et al., 2011a*; *Zheng et al., 2011b*). In contrast with AMS3 lipase, the enzyme was stable from acidic to alkaline condition. These properties make the enzyme more valuable in detergent and oleochemical industries. Enzymes are proteinaceous, meaning that they have amino acids whose ionization can be affected by pH. The primary and secondary structures of the enzymes can be affected, thereby affecting the activity. The primary and secondary structures of the AMS3 lipase may be influencing these pH and temperature stabilities, leading to the variation in the ionization of the native AMS3 lipase enzyme at different pH.

### Effect of metal ions

The effect of metal ions was studied to determine which metal ions could enhance or reduce the activity of AMS3 lipase. All of the metal ions studied enhanced lipase activity at 1 and 5

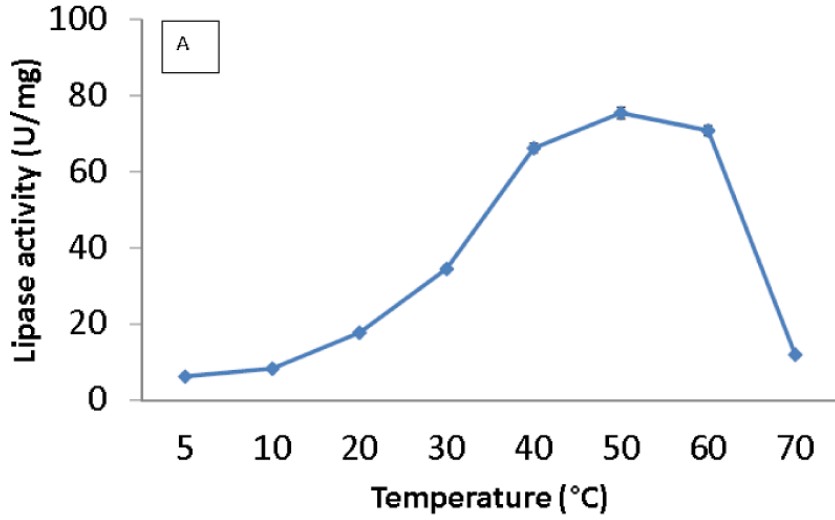

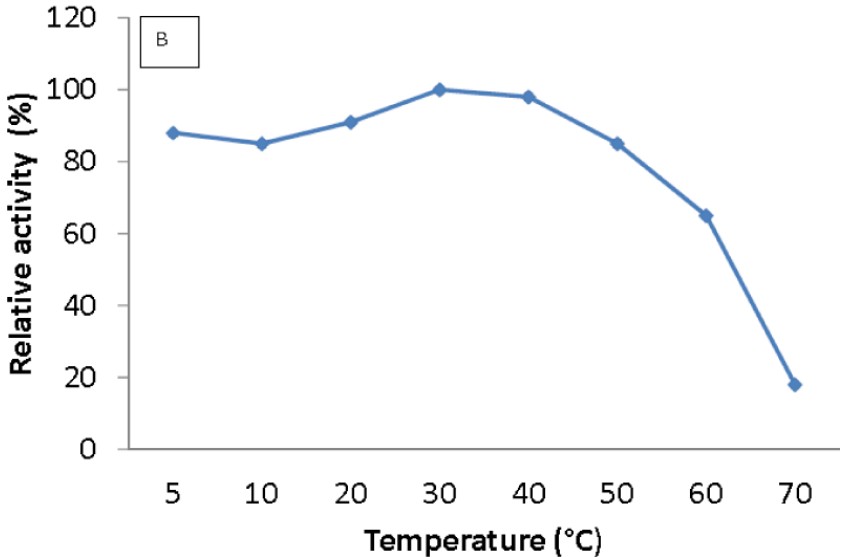

**Figure 3 Effect of temperature on purified AMS3 lipase activity.** The lipase assay was done using olive oil as a substrate. (A). Enzyme samples were assayed at various temperatures as mentioned in the Methods section. The temperature that showed the highest lipase activity was considered to be the optimum temperature for the AMS3 lipase. The effect of temperature on purified AMS3 lipase stability (B). Enzyme stability was tested at different temperature. The enzyme was incubated at various temperatures in 30 min. The assay was performed at optimum temperature. The relative activity was calculated using 30 °C as 100%.

mM (Fig. 5). For $Li^+$ and $Rb^+$ lipase activity was heavily enhanced, with relative activities over then 160%. $Li^+$ have been reported to improve the enantioselectivity of the lipase (Okamoto et al., 2006). $Na^+$ and $K^+$ were also found to enhance AMS3 lipase activity. Metal ion activation of enzymes is important in industrial applications for obtaining maximal catalytic efficiency. However, at 5 mM concentration, $Li^+$, $Ni^{2+}$ and $Zn^{2+}$ reduced lipase activity to less than 50%. In contrast, *Sharma et al. (2002)* reported that $Ca^{2+}$ ions increased

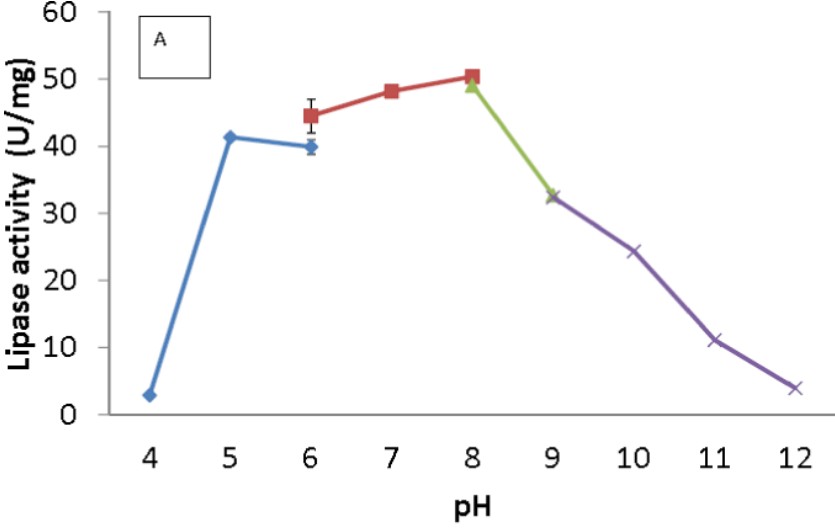

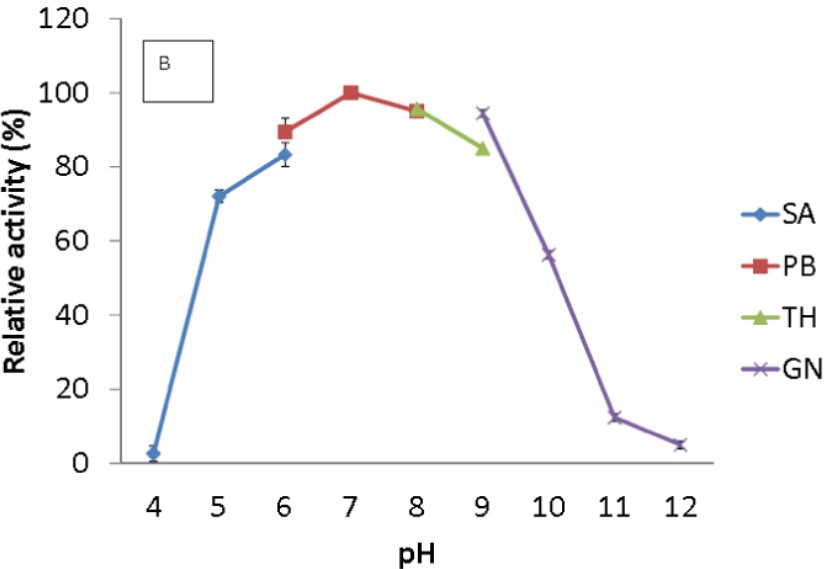

**Figure 4  Effect of pH on purified AMS3 lipase activity.** (A) The lipase activity was measured at different pH using olive oil emulsion as substrate and assay at 50 °C for 30 min. Effect of pH on purified AMS3 lipase stability is shown in figure (B). The enzyme was pre-incubated at different buffer for 30 min. The lipase assay was performed at 50 °C. Sodium acetate buffer was used for the pH range of 4.0 to 6.0 (♦); potassium phosphate buffer for the pH range of 6.0 to 8.0 (■); Tris-HCl buffer for the pH range of 8.0 to 9.0 (▲); and glycine-NaOH buffer for the pH range of 9.0 to 12.0. (x). The relative activity was measure using pH 7 as 100%.

the lipase activity of *Bacillus* sp. RSJ-1 by 16%, similar to the lipase activity of *B. coagulans* MTCC-6375 (*Sharma et al., 2002*).

## Substrate specificity

Assays addressing the specificity of the AMS3 lipase towards natural oils were conducted. The results showed that the AMS3 lipase hydrolyzed all of the natural oils tested, and

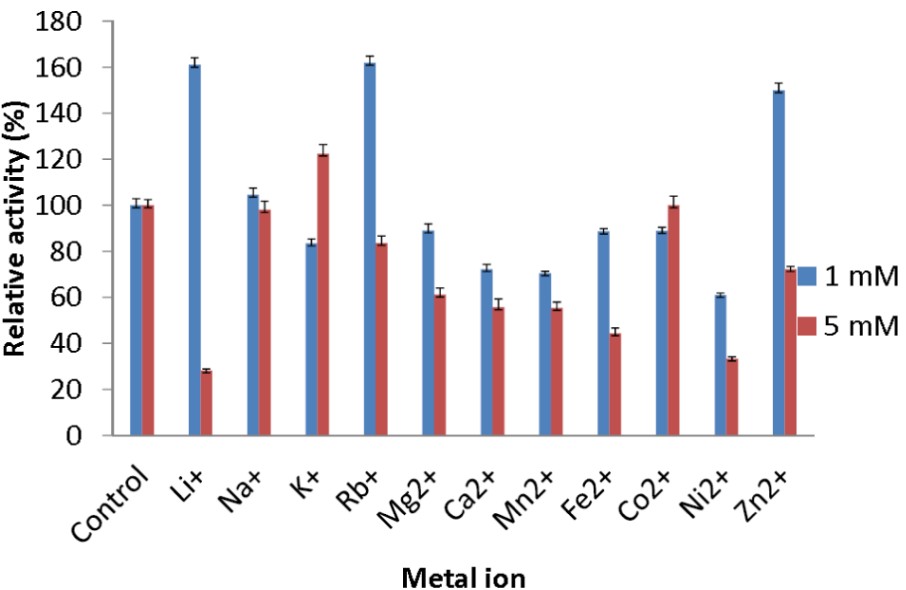

**Figure 5** **Effect of metal ions on AMS3 lipase stability.** All assay was done using olive oil as substrate. The enzyme was pre-incubated at 50 °C with various metal ions at concentrations of 1 mM (blue) and 5 mM (red). The treated enzyme was assay at 50 °C for 30 min. The untreated enzyme was considered as a control or 100%.

the highest activity was seen with sunflower oil (Fig. 6). Olive oil is composed of mixed triglycerides (monounsaturated C18:1). Most lipases show a preference for long-chain fatty acids. The extracellular lipase Lip2 from *Yarrowia lipolytica* showed a much higher lipolytic activity toward triglycerides in olive oil and safflower oil than toward hydrophilic esters such as methyl fatty acid asters (*Jaeger, Dijkstra & Reetz, 1999*).

The AMS3 lipase displayed the second highest activity with coconut oil, with a relative activity of 110%, followed by canola oil, olive oil and sesame oil with relative activities of 106%, 100% and 90%, respectively. Similarly, AMS3 lipase exhibited the same substrate specificity with an extracellular lipase from *Pseudomonas sp* S5 (*Rahman et al., 2005*). In contrast, a *Pseudomonas aeruginosa* EF2 lipase reported olive oil as its preferred substrate (*Gilbert, Cornish & Jones, 1991*). AMS3 lipase has a broader substrate specificity ranging from C12 to C18. The ability of AMS3 lipase to hydrolyze long carbon chain triglycerides (C12 above) showed that the enzyme is a true lipase (Jaeger et al., 1994). According to Jaeger et al. (1994), surface pressure and lipid distribution are physical factors that can impact the determinants of lipase activity (*Hiol et al., 1999*). Therefore, as reported by *Hiol et al. (1999)*, the differences in substrate specificity can arise not only from specific substrate binding but also from differential activation of the enzyme at the interface (*Lanser, Manthey & Hou, 2001*).

### Effect of organic solvents

*Persson et al. (2002)* reported that the use of organic solvents in enzymatic catalysis is now well recognized with several advantages in water-poor mediums including improved solubility of hydrophobic substrates, a shift in thermodynamic equilibrium to synthesis

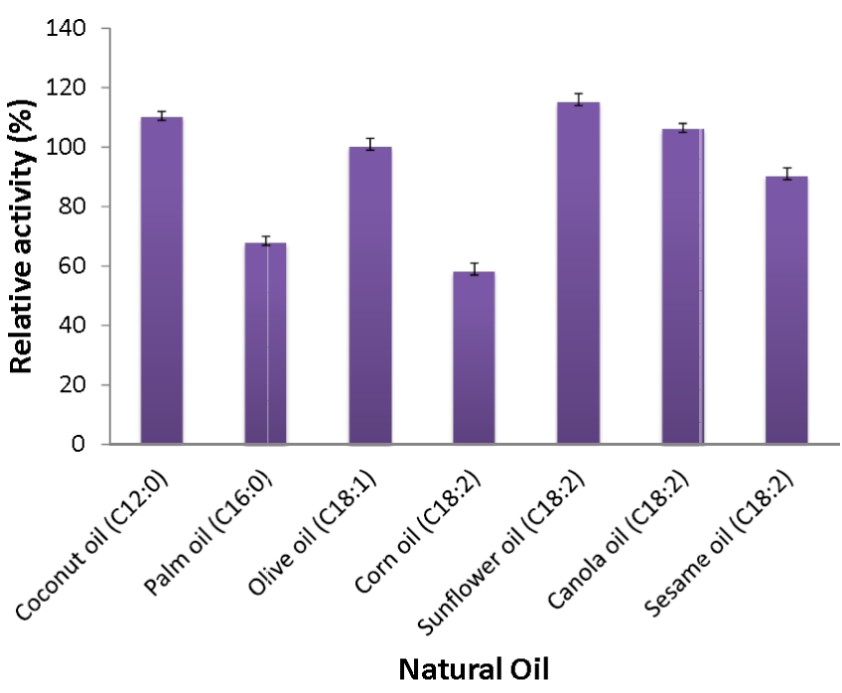

**Figure 6  Substrate specificity of purified AMS3 lipase toward natural oil.** The assay was performed at 50 °C for 30 min. Substrate emulsion prepared with various natural oil with ratio 1:1 (buffer and natural oil). Data shown are the means from triplicate measurements expressing % activity relative to that with olive oil.

from hydrolysis and increased thermostability of the enzyme (*Persson et al., 2002*). Assays measuring the effects of solvents were conducted to understand AMS3 lipase activity in organic solvents. The enzyme was incubated in a 25% v/v solvent at a temperature of 50 °C (Fig. 7). The results showed that AMS3 lipase exhibited some enhancements in stability expressed as relative activity in 25% v/v solvent compared to the control (without solvent). For instance, AMS3 lipase has a relative activity of 74%, 89% and 116% in DMSO, methanol and n-hexane, respectively, compared with its original activity at a temperature of 50 °C. The stability in the said organic solvents makes AMS3 lipase a good candidate for synthesis of biodiesel and transesterification reactions. Lipases have a sheath of water molecules tightly bound to them. This sheath protects the enzyme's hydrophilic surface and possibly retains the native conformation even in non-polar solvents. Moreover, the changes in enzyme stability in organic solvents are also greatly affected by the structure of the enzymes (*Gomez-puyou, 1992*).

## Secondary structure analysis of AMS3 lipase

As illustrated in Fig. 8A, the resulting denaturation (sigmoidal graph shape) curve indicated a structural transition of AMS3 lipase within the tested temperature range. The $T_m$ value for AMS3 lipase for this transition was 62 °C (Fig. 8A). The Tm measurement using circular dichroism is close to the AMS3 lipase deactivation profile, which was at 55 °C. CD measurements have been widely used to follow the equilibrium between helical structures and unordered conformations. The CD spectra (molecular ellipticity) of AMS3 lipase was

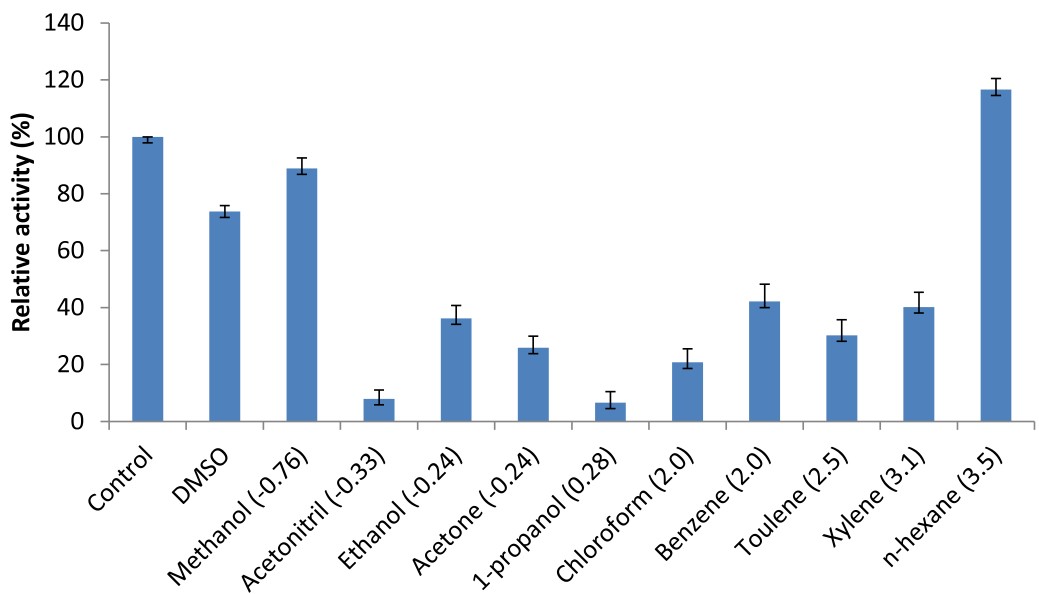

**Figure 7** **Effect of solvents on purified AMS3 lipase.** All the assay was done using olive oil as a substrate. Incubation time for treated enzyme with organic solvent was 30 min and assay at 50 °C. Remaining lipase activity was expressed relative to that of the control (without solvent). Log P (water/octanol coefficient) value of each solvents are represented in parenthesis.

analyzed as a function of temperature between 190 and 260 nm. The wavelength 220 nm was set to monitor the transition of $\alpha$-helical to unordered structures as they exhibited characteristic signals at this wavelength. The high-tension voltage (HT) value gradually increased until the $T_m$ of AMS3 lipase and then decreased as the lipase unfolded above the melting temperature point. This outcome might be due to loss of protein secondary structure, followed by an increase in unordered conformations. This observation was supported by the secondary structure content determination at different temperatures. As depicted in Fig. 8B the increased formation of loop structures or disordered regions were observed notably at temperatures from 50–90 °C. In fact, AMS3 lipase exhibited a decreasing content of the alpha helices. This behavior could turn out to be the deactivation factor for AMS3 lipase. The enzyme gradually lost its activity beyond 50 °C. The random secondary structure conformation appears to be increasing as temperature drops together with $\beta$-pleat contents. The increased loop confirmation was found to be one of the factors for enzymes to work at low temperatures (*Amico et al., 2002*). These findings could explain the AMS3 lipase broad temperature profile.

## CONCLUSION

The AMS3 lipase was successfully purified via single-step affinity chromatography. The purified AMS3 lipase exhibited optimum lipolytic activity at pH 8 with a broad temperature profile from 10–70 °C which is unique for an enzyme isolated from Antarctic microorganism. In addition, the enzyme had shown to be stable when treated with different buffer condition ranging from pH 5–10. This factor will make the enzyme very useful for

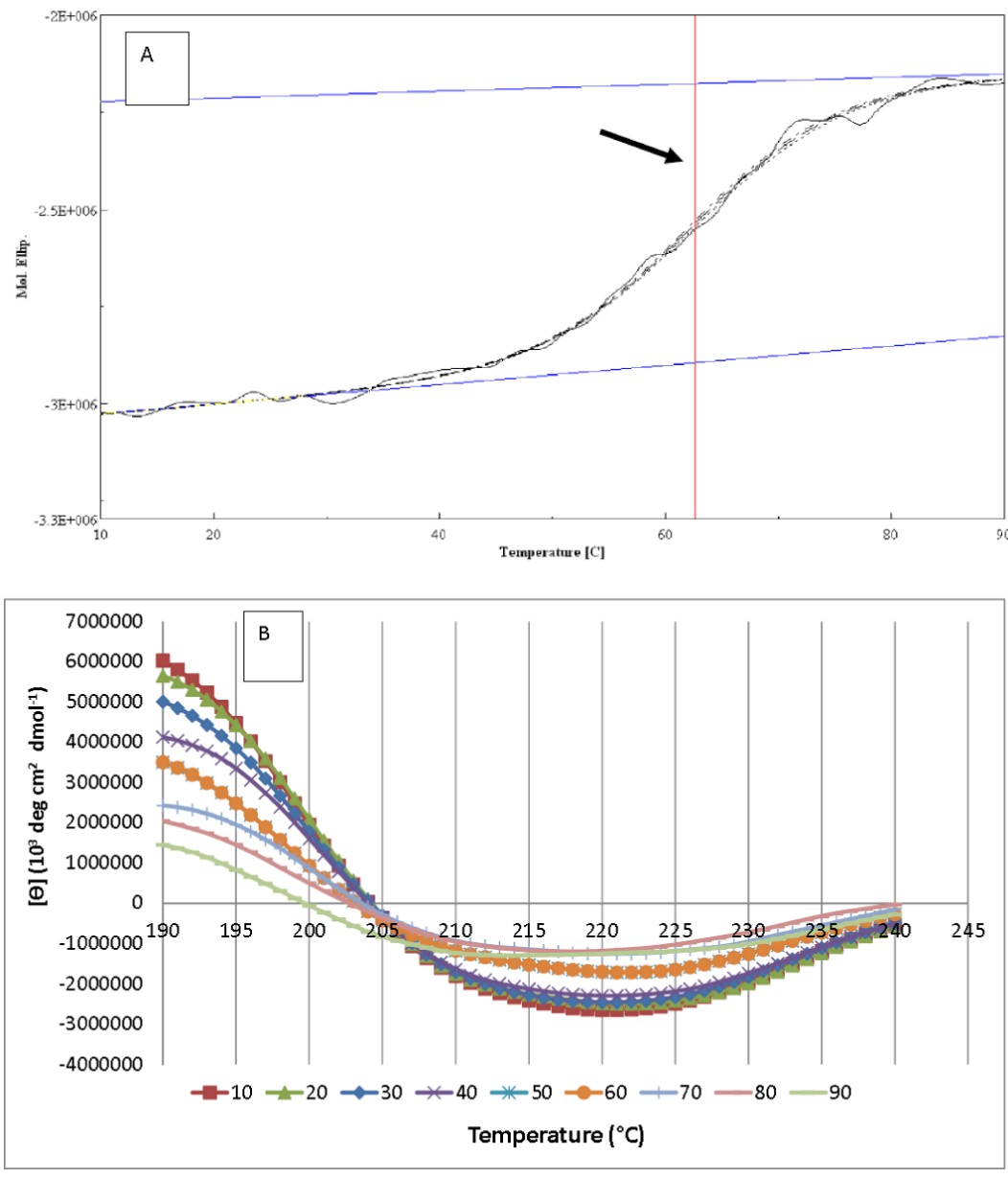

**Figure 8** **AMS3 lipase melting point and secondary content determination.** Vertical line (arrow) indicating the melting point value of AMS3 lipase when tested from temperature ranging from 10–90 °C (A). Changes on the secondary structure content of AMS3 lipase were notably observed as the enzyme was subjected at different temperature (B).

detergent industry. As shown with other lipases, the AMS3 lipase efficiently hydrolyzed unsaturated fatty acids and sunflower oil with the highest activity. Various divalent and monovalent metal ions increased the catalytic activity of the AMS3 lipase, but $Ni^{2+}$ completely decreased its activity. Additionally, the AMS3 lipase exhibited stability in 25% v/v of n-hexane, methanol and DMSO at 50 °C. These unique properties are believed to provide tremendous economic benefits in pharmaceuticals, cosmetics and synthesis of functional lipids as well as other low-high temperature industrial applications.

### Funding

This research was supported by research grant University Putra Malaysia. GP-IPS/2016/9479200. The funders had no role in study design, data collection and analysis, decision to publish, or preparation of the manuscript.

### Grant Disclosures

The following grant information was disclosed by the authors:
University Putra Malaysia: GP-IPS/2016/9479200.

### Competing Interests

The authors declare there are no competing interests.

### Author Contributions

- Wahhida Latip conceived and designed the experiments, performed the experiments, analyzed the data, wrote the paper, prepared figures and/or tables, reviewed drafts of the paper.
- Raja Noor Zaliha Raja Abd Rahman and Mohd Shukuri Mohamad Ali conceived and designed the experiments, analyzed the data, contributed reagents/materials/analysis tools, reviewed drafts of the paper.
- Adam Thean Chor Leow and Fairolniza Mohd Shariff analyzed the data, contributed reagents/materials/analysis tools.

### DNA Deposition

The following information was supplied regarding the deposition of DNA sequences:
Genbank accession number KP744536 and KT215165.

### Data Availability

GST C Pseudomonas A3 (KT215165), Lipase AMS3 (KP744536).

### Supplemental Information

Supplemental information for this article can be found online at http://dx.doi.org/10.7717/peerj.2420#supplemental-information.

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
