# Peer review of "Expression and characterization of thermotolerant lipase with broad pH profiles isolated from an Antarctic Pseudomonas sp strain AMS3"

_PeerJ, doi:10.7717/peerj.2420_

## Round 0.1 · original submission · Major Revisions

· Academic Editor

Major Revisions

I agree with all points made by both reviewers:

Please revise carefully all figures and provide them with sufficient resolution, and self explaining legends.

The article should include a clear introduction and background to demonstrate how the work fits into the broader field of knowledge. Please check again the relevant literature and reference all previous work that helps to understand the background and context of your work.

Concerning the experiment performed, please describe carefully how many independent experiments were performed and which statistically analysis was used.

Another final round of English editing would be useful as well.

Reviewer 1 ·

Basic reporting

Figures should be relevant to the content of the article, of sufficient resolution, and appropriately described and labeled.

The article should include sufficient introduction and background to demonstrate how the work fits into the broader field of knowledge. Relevant prior literature should be appropriately referenced.

Experimental design

Methods should be described with sufficient information to be reproducible by another investigator

Validity of the findings

The data should be robust, statistically sound, and controlled.

Additional comments

In this work, the authors have cloned and expressed the gene encoding a thermotolerant lipase with broad pH from an Antarctic Pseudomonas strain AMS3. The recombinant lipase AMS3 was purified by single-step purification using affinity chromatography. The effect of organic solvants and ions on this enzyme were studied and the authors showed that various mono and divalent metal ions increased the activity of the AMS3 lipase. In addition, the enzyme activity in 25 % v/v solvents at 50 °C particularly to nhexane, DMSO and methanol could be useful for catalysis reaction in organic solvent and at broad temperature.
The paper is well written and experiments conducted properly. However, in my opinion, this work requires significant revisions which I feel are major enough that I would need to re-evaluate any revised version.

* Line 82: replace "Identififcation" by "identification".
* line 83: replace "were send" by "were sent".
* Line 84: "under accession no was": please delete "was".
* Lines 84 to 86: The authors should give more details for screening lipase activity (show the role of tributryin as substrate).
* Line 91: olive oil can not be emulsified by aqueous buffer. The authors should use arabic gum as emulsifier.
* Line 92: the hydrolysis of olive oil as substrate generete long chain fatty acids which can affect the lipase activity. So, authors should decreased the time of incubation for reaction mixture (30 min is too much).
* Line 95: replace "absorbency" by "absorbance".
* Materials and methods section: The primers used in this work should be added (forword and reverse).
* Why lipase was expressed as tagged (His-tagged)? In addition, tagging can affect the structure and the activity of the native form of this lipase?
* Line 171: in my opinion, phosphate does not buffers at pH 8. Tris-HCl buffer is better between pH 7.5 and 8.5.
* After each incubation in various temperatures and pH, samples must be centrifuged to eliminate denaturating proteins.
* Section: 2.7.5: in my opinion apolar solvant like hexan and chloroform were immiscible in buffer containing enzyme. So, we can not evalute their effect on the purified lipase.
* Line 210: replace "mints" by "min".
* Line 208: replace "n1-3" by "sn1-3".
* Line 226: can authors explain why they have used 222 nm as wave length for DC study.
* Line 264: replace "Is" by "is".
* Line 266: replace "section respectively" by "sections, respectively.
* Line 268: replace "Multiople" by "multiple".
* Line 270: "between known lipase from various microorganism lipase exhibited". Please deleted the second "lipase".
replace "conserve" by conserved".
* Line 273: replace "includes" by "including".
* Line 279: replace "for Pseudomonas" by from Pseudomonas".
* Line 281: "to fold to". Please delete the second "to".
* In fig. 1 legend: @ is not His in the fig.1. Colored residues indicate what?
* Line 308: authors should add ref. Lip2.
* Line 313: replace "Stphylococcus" by Staphylococcal.
* Fig. 2: the band of lipase is not clear. Authors should increase the quantity of enzyme loaded in the acrylamide gel.
What indicate piste 4 in the fig.2
* Line 330: make "however" between two comma.
* Line 356: replace "with stabilities ranging from pH 6 to 9" by "and stable at pH ranging from 6 to 9".
* Line 359: make "whereas" between two comma.
* Line 455: replace "fig. 5A by "fig. 9A".
* Line 468: the sentence is not clear. In my opinion, authors should replace "in contrast" by "in fact".
* Fig. 8: the authors should add chloroform/acetone percentage in the legend. In addition, I noticed that hydrolysis of triolein is very weak. So, authors should further optimize the hydrolysis conditions.

Reviewer 2 ·

Basic reporting

Figures should be relevant to the content of the article, of sufficient resolution, and appropriately described and labeled.

The submission should be ‘self-contained,’ should represent an appropriate ‘unit of publication’, and should include all results relevant to the hypothesis. Coherent bodies of work should not be inappropriately subdivided merely to increase publication count.

Experimental design

The submission should clearly define the research question, which must be relevant and meaningful. The knowledge gap being investigated should be identified, and statements should be made as to how the study contributes to filling that gap.
The investigation must have been conducted rigorously and to a high technical standard.
Methods should be described with sufficient information to be reproducible by another investigator.
The research must have been conducted in conformity with the prevailing ethical standards in the field.

Validity of the findings

The data on which the conclusions are based must be provided or made available in an acceptable discipline-specific repository.
The conclusions should be appropriately stated, should be connected to the original question investigated, and should be limited to those supported by the results.
Speculation is welcomed, but should be identified as such.

Additional comments

The manuscript entitled “Expression and characterization of 1,3 positional specificity thermotolerant lipase with broad pH profiles isolated from an Antarctic Pseudomonas sp strain AMS3” present results of biochemical characterization of a new lipase, however, there are many points that can be improve before the manuscript can considered as an original article in Peer J.

Major:

1. In general, the materials and methods used are not described properly.
- Line 88: the standard method of lipase activity should be described here (T and pH), in line 92 appear that the assay was perform at 4 °C, this is valid? Please clarify
- Line 160: the kind of markers should be report here.
- Line 166, 2.7.1 Effect of temperature section: For thermal stability assays, the time of enzyme incubation at each temperature before perform the test of activity should written; with these values, an experiment to know the half-live can be calculated.
- Line 207, 2.7.7 Positional specificity of recombinant AMS3 lipase section: in lines 211-212 the proportion of solvents and the time of equilibration in the chamber before perform the TLC should be written; the quantity (g) of deposed of standards and triolein hydrolysate into TLC should be mentioned. In general, the spots of triolein and their glyceryl species not are visible just like that; a derivatization or exposition to one revelator is perform. Please complete this methodology.
2. Line 244: please clarify the use of “true” lipases, because lipases are able to hydrolyze several substrates, as porcine pancreatic lipase that can hydrolyze triolein but has a better specific activity when tributyrin is use as substrate.
3. Figure 1: is not clear the use of @ for catalytic Histidine (H); the @ is up of an aspartic acid (D), please clarify and correct this point.
4. Line 304: the protein purification fold of 6.84 do not coincide with the fold reported in the purification table from raw data file.
5. Lines 304-310: which it is what the authors mean by this? Sometimes also, the recombinant enzymes need two or more purification step to have a clear purified enzyme; in my opinion, these lines can be delete. Please clarify
6. Figure 2. It is strange that having such a high performance and a significant amount of pure enzyme, the gel shown for pure protein is diffuse and unclear (line 3), apparently, not enough protein was loaded. The gel should be replace with one where a clear band is show and indicate the amount of protein added (10-20 g). Thus would be sufficient to demonstrate a purified protein if is the case. Also, add the units of numbers presented for markers.
7. Figure 3: the symbol presented for °C is incorrect, change ‘ by °. The lipase activity should be presented as U/mg not as U/mL (Fig. 3A). This Figure should be change. Increase the size of symbols used. Line 348: please rewrite this sentence indicating the time of incubation and conditions for thermal stability assays including the substrate. Also the conditions to measure the remained activity or indicate a material and methods sections. The % should be presented as (%).
8. Figure 4: in Fig. 4A the lipase activity should be presented as U/mg not as U/mL. This Figure should be change. Please indicate the time of incubation and conditions for thermal stability assays including the substrate. Also the conditions to measure the remained activity or indicate a material and methods sections. Increase the size of symbols used. A blue diamond point at pH 10 is presented at 0% of relative activity, please clarify. The % should be presented as (%).
9. Figure 5: Separate the value of 1 or 5 form mM in the legend and the Figure legend. The % should be presented as (%). Lines 389-390: please indicate the conditions used as suggested in Figures 3 and 4 (e.g. time, pH of incubation, etc).
10. Figure 6: The % should be presented as (%). Please indicate the conditions used as suggested in Figures 3 and 4.
11. Lines 432-433: A reference should be added.
12. Figure 7: is not necessary to add (100%) in the Y axis, only (%) should be written.
13. That is what the authors mean by sn 1-2 ? They mean sn-2 ? Please clarify.
14. Figure 8: The TLC presented do not have the quality for regioselectivity determination. In fact, this experiment should be repeat and quantified if it is possible. Also in the Figure text the supplier of standards used should be added. An example of a good TLC is showed in the follow link: https://www.google.com.mx/search?tbm=isch&q=tlc+for+lipase+regioselectivity&hl=es&#imgrc=lr6vG0mKqxPaPM%3A
If these experiments are not repeat to obtain a reliable result, this section do not be included in the manuscript and all sections related with regioselectivity should be delete.
15. Finally, the title have some particular characteristics of a new lipase; however, all the results don’t support this point; therefore , the title should be re-written according the results presented; an example can be “Expression and characterization of thermotolerant lipase with broad pH profiles isolated from an Antarctic Pseudomonas sp strain AMS3”.


Minor:

1. Line 82: use a space after “.”; this kind of error is very repetitive in all the manuscript.
2. Line 84: add a “.” after (KR821141) or sp.; this kind of error is very repetitive in all the manuscript.
3. Line 94: Indicate the pH of acetate pyridine solution
4. Lines 96- 97: the definition of lipase unit it not correct and must be written in past.
5. Write as the SI units indicates, e.g. 40°C, “°C” is written with one space after the number, 40 °C; this kind of error is in the whole manuscript.
6. Line 168: when several numbers have the same unit, the SI unit is added after the last one, always whit a space between.
7. Line 174: Why use the term mutant lipase, in this work, the mutagenesis was perform? Please clarify.
8. Line 92, 189, 211: use only notation of SI for units; sometimes minutes, mints or min is used, please use min in all the manuscript; also h instead hour, etc.
9. Line 210: change n-hexane instead n-haxane
10. Line 221: use a space after the value of one unit, 0.1 mg/ml instead 0.1mg/ml. This kind of error is very repetitive in all the manuscript.
11. Line 287: use the special format of cursives for scientific names.
12. Line 442: write sn-1 and sn-3 instead sn1 and sn3
13. References: the scientific names should be written in cursive text. There are many corrections to do in this section: e.g. references: 1,4,7,8, 10, 19, 21, 25-28, 32-35.

---

## Round 0.2 · Major Revisions

· Academic Editor

Major Revisions

Please revise your manuscript carefully again according to the suggestions of reviewer #2.

Reviewer 1 ·

Basic reporting

The submission must adhere to all PeerJ policies.

Experimental design

Methods should be described with sufficient information to be reproducible by another investigator.

Validity of the findings

The conclusions should be appropriately stated, should be connected to the original question investigated, and should be limited to those supported by the results.

Reviewer 2 ·

Basic reporting

Figures should be relevant to the content of the article, of sufficient resolution, and appropriately described and labeled.

The submission should be ‘self-contained,’ should represent an appropriate ‘unit of publication’, and should include all results relevant to the hypothesis. Coherent bodies of work should not be inappropriately subdivided merely to increase publication count.

Experimental design

The submission should clearly define the research question, which must be relevant and meaningful. The knowledge gap being investigated should be identified, and statements should be made as to how the study contributes to filling that gap.

Validity of the findings

The conclusions should be appropriately stated, should be connected to the original question investigated, and should be limited to those supported by the results.

Additional comments

The manuscript entitled " Expression and characterization of thermotolerant lipase with broad pH profiles isolated from an Antarctic Pseudomonas sp strain AMS3” was corrected partially according to reviewer questions; this manuscript present results of biochemical characterization of a new lipase, however, there are some points to improve before the manuscript can considered as an original article in PeerJ.Major:
1. The feature of this new enzyme is a thermotolerant lipase with a broad pH profile, however, the specific activity on olive oil is and other oils is lower compared to other lipases from Pseudomonas (e.g. Lipase from Pseudomonas aeruginosa: 6600 U/mg with olive oil, Gilbert et al., 1991). Please justify Please justify in the manuscript the advantage of this enzyme compared to other lipases from Pseudomonas despite having low activity with oils.
2. Other lipases that can act in these ranges of pH and temperature (e.g. Lipase of Thermomyces lanuginosus, Lipase A of Candida antarctica, …). Please justify in the manuscript the advantage of this enzyme compared to other lipases.
3. Figure 2. The experiment charging a quantity between 10-20 ug of protein was done as suggested by the reviewer; the lane of purified enzyme shows a major band and also several bands corresponding to other proteins, that means that the lipase is not in a pure form, is only enriched. This fact must be taken into account and the authors must change the term purified for enriched in this Figure and in the entire manuscript including the abstract.
4. Figure 2: the native migration in of “purified enzyme” (lane 4) does not provide information in purity. Please justify it inclusion in this Figure.
5. Why the maximum specific activity (U/mg) obtained in Figures 3 and 4 (is minor to the obtained in the purification Table presented in raw data (139 U/mg); please clarify
6. In raw data: why all the experiments for each condition have the same error % (see in raw data: Effect of temperature on lipase activity, Effect of temperature on stability, Effect of pH on activity, Effect of pH stability, Effect of metal ion)? Why the value of each standard deviation is included instead the % of error; is strange to have exactly the same % of error in each kind of experiments, this is not normal in experimental procedures and appears that these error values were assigned instead obtained. Please clarify
7. From graphics 3 and 4, can be observed that this values not coincides with the presented in raw data; e.g.: 50 °C from Table 75.41±2% (75.41±1.51), from Figure 3 about 75±5 (about 6% of error), the same of the others temperatures and pH. Please clarify
8. Figure 6 is wrong presented and must be changed: in Y-axis, the title must be natural oils instead relative activity; relative activity must be added to X.
9. Figure 8b: the title of Y-axis must be added, in X axis remove °C from labels because is not included in all temperatures and add °C in X-axis.
Minor:
• Figures 3 and 4: place U/mg between parentheses (U/mg)

---

## Round 0.3 · Minor Revisions

· Academic Editor

Minor Revisions

- Although figures 3 and 4 have been improved, it still remains a problem that the maximal lipase activities differ quite remarkably. Please explain that at least in the corresponding text.

- Please revise your manuscript one more time and compare as suggested by reviewer 2 previously your enzyme activities with those of existing enzymes.

- Please focus in your conclusions on what is new and specific for the lipase you describe in this publication.

---

## Round 0.4 · accepted · Accept

· Academic Editor

Accept

Thank you very much for your careful revisions.